# Profibrotic Inflammatory Cytokines and Growth Factors Are Predicted as the Key Targets of *Uncaria gambir* (Hunter) Roxb. in Keloids: An Epistatic and Molecular Simulation Approach

**DOI:** 10.3390/ph17060662

**Published:** 2024-05-21

**Authors:** Sri Suciati Ningsih, Fadilah Fadilah, Sri Widia A. Jusman, Rahimi Syaidah, Takashi Yashiro

**Affiliations:** 1Doctoral Program in Biomedical Sciences, Faculty of Medicine, Universitas Indonesia, Jakarta 10430, Indonesia; sri.suciati11@ui.ac.id (S.S.N.); fadilah.msi@ui.ac.id (F.F.); 2Faculty of Medicine, Universitas Muhammadiyah Prof. Dr. Hamka, Jakarta 12130, Indonesia; 3Department of Medical Chemistry, Faculty of Medicine, Universitas Indonesia, Jakarta 10430, Indonesia; 4Department of Biochemistry, Faculty of Medicine, Universitas Indonesia, Jakarta 10430, Indonesia; sriwidiaaj@gmail.com; 5Center of Hypoxia and Oxidative Stress Studies, Faculty of Medicine, Universitas Indonesia, Jakarta 10430, Indonesia; 6Department of Histology, Faculty of Medicine, Universitas Indonesia, Jakarta 10430, Indonesia; 7Department of Histology, Jichi Medical University School of Medicine, Tochigi 329-0498, Japan; tyashiro@jichi.ac.jp

**Keywords:** gambir, keloid, target protein, molecular simulation

## Abstract

Keloid is characterized as the fibrotic tissue resulting from the increase of fibroblast activity. *Uncaria gambir* (Hunter) Roxb. possesses bioactive compounds that have potential as antifibrotic agents, while the mechanism of action in keloid has not yet been elucidated. The aim of this study was to investigate the interaction of gambir bioactive compounds with keloid target proteins using an epistatic and molecular simulation approach. The known bioactive compounds of gambir targets and keloid-related protein targets were screened using databases. The network was constructed and analyzed to obtain the core protein targets. The targets were enriched to describe the Gene Ontology (GO) and pathway related to the proteins. Eleven targets were defined as the main targets of gambir bioactive compounds related to keloid disease. Gambiriin C, Isogambirine, and Procyanidin B1 were identified as the most promising compounds with the highest binding energy to transforming growth factor beta 1 (TGFβ1), AKT serine/threonine kinase 1 (AKT1), and matrix metallopeptidase 1 (MMP1) as the target proteins. GO enrichment and pathway analysis found that gambir bioactive compounds may act on keloid-related target proteins to regulate cell proliferation, migration, transcription, and signal transduction activity via profibrotic cytokine and growth factor signaling pathways. This study provides a reference for potential targets, compounds, and pathways to explain the mechanism of gambir against keloid.

## 1. Introduction

Keloids display exophytic overgrowth beyond the site of injury. The latest demographic study of keloid incidence reported that the incidence of keloids is higher in pigmented-skin individuals, particularly in Africa (8.5–16%). Even though this disease may not directly result in fatality, the manifestation may cause cosmetic, physical, and physiological effects that consequently affect the patient’s quality of life [1,2]. The pathomechanism of keloid has not been fully elucidated. Abnormality of the wound-healing process is addressed as the critical point of keloid formation. Robust inflammatory stimulation and over-sensitized fibroblasts driven by cytokines and growth factors during the wound-healing process contribute to excessive extracellular matrix (ECM) deposits in keloids. Over-sensitized fibroblasts are activated into myofibroblasts, which tend to have a higher level of proliferation, migration, and extracellular matrix synthesis capacity [3,4,5].

Several existing treatments for keloid, including surgery, steroid therapy, stem cell, radiation, and targeted therapy, are intended to inhibit the proliferation and activation of keloid fibroblasts. Unfortunately, none of these are considered a universal gold standard for keloid treatment due to the high recurrence rate, emergence of side effects, and requiring high cost and further study to reveal the effects in a clinical approach [2,6,7]. Thus, herb-based therapy is proposed as an option for an alternative way to treat keloid disease. Traditional herbal medicine possesses the characteristics of being multicomponent, multifunction, and multitarget; is of low toxicity; and has limited side effects, in general [8].

Gambir is an herbal medicine originating from Indonesia that has long been used by local residents as a traditional remedy. A majority of the phytochemical substances derived from gambir are categorized into the flavonoid, phenolic, and alkaloid groups [9]. Recent studies proved that the bioactive compounds of gambir have antibacterial, anticancer, anti-inflammation, as well as antifibrotic effects [10,11,12,13]. Desdiani et al. [14,15] found using in vivo study that gambir has antifibrotic effect against lung fibrosis through inhibition of nuclear factor kappa-B (NF-κB) and transforming growth factor receptor (TGF-βR). In line with these findings, Sriningsih [16] identified that gambir extract could inhibit the expression of tissue inhibitor matrix metalloproteinase (TIMP) and collagen-1 in rat liver fibrosis tissue. Keloid is defined as tissue fibrosis due to its effusive accumulation of ECM, particularly collagen [2]. Jusman et al. [17] have shown that some gambir bioactive compounds have potency to inhibit platelet-derived growth factor subunit A (PDGFA) as a target protein to reduce fibroblast proliferation through in silico study. Therefore, gambir is assumed to be a promising herb-based antifibrotic agent to deal with keloid disease.

Recent studies have highlighted the efficiency of bioinformatic analysis as a beneficial tool to explore the molecular mechanism of disease [18]; one form of this is epistatic study. Epistatic study is a beneficial tool to discover a new drug candidate for human disease by understanding the gene interaction and pathway that contribute to the disease risk. Furthermore, the researcher can identify potential targets that can disrupt the interaction and reduce the disease progression [19,20,21]. One branch of epistatic interaction study is network pharmacology, which can be used as an appropriate approach to analyze and model the interactions between drug–target networks [22]. To date, a comprehensive understanding of gambir’s mechanism of action against fibrotic tissue, particularly keloid, is still lacking. The conventional experimental method to explore all the proteins included in the mechanism is time-consuming, expensive, and so forth [23]. Therefore, this bioinformatic approach is suggested as an efficient pathway to analyze the interactions between keloid-related proteins and the bioactive compounds of gambir. This study aims to investigate the bioactive compounds of gambir and their interaction with keloid target proteins using a network pharmacology and molecular simulation approach. The result of this study is expected to provide a scientific explanation of the mechanism of gambir as an antikeloid candidate.

## 2. Results

### 2.1. Screening for Uncaria Gambir Bioactive Compounds and Its Target Proteins

A total 74 bioactive compounds of *Uncaria gambir* (Hunter) Roxb. were identified from the literature study [9,17,24], with 5048 protein-specific targets based on SWISS target prediction. A total of 277 targets from 29 bioactive compounds were predicted to be targeted to keloid-related target proteins. The drug-likeness nature, pharmacokinetic properties, and toxicity of the compounds are shown in Table 1. The drug-likeness properties identified from this study were the molecular weight (Mw), octanol–water partition coefficients (Log P), hydrogen-bond acceptor, and donor (HBA and HBD). The pharmacokinetic parameters included were the gastrointestinal absorption (GI Abs) and cytochrome (CYP) inhibition. The toxicity of the compounds was described by the toxicity class (1–6) according to the globally harmonized system of classification for the labeling of chemicals (GHS). Classes 1–2 mean fatal if swallowed, Class 3 is toxic if swallowed, Class 4 is harmful if swallowed, Class 5 may be harmful if swallowed, and Class 6 is non-toxic. All of the compounds were categorized in Classes 3 to 6.

### 2.2. Screening for Keloid-Related Target Proteins

A total of 96 out of 695 keloid-related target proteins were identified after combining the 2 databases. The 96 genes were then imported into Cytoscape 3.9.1 to plot the keloid-related target network diagram. A total of 96 nodes and 100 edges were obtained using the Cytoscape 3.9.1 platform, with a high confidence score (0.98). The top 11 of the 96 targets were selected based on the degree, betweenness centrality (BC), and closeness centrality (CC) score (Table 2). The keloid-related protein–protein interaction (PPI) network was visualized using GeneMANIA (Figure 1).

### 2.3. Interaction of Bioactive Compounds of Gambir on Keloid-Related Target Proteins

A total of 277 targets from 29 bioactive compounds were predicted to be targeted to keloid-related target proteins. All of the gambir bioactive compounds’ targets were merged with the keloid-related targets and keloid-related targets with a high degree value (HDV) to further discover the potential targets and mechanism (Figure 2A). Based on the diagram, there were 11 target proteins defined in the intersection area. All of these target proteins were defined as the main targets of gambir bioactive compounds in treating keloid disease. The drug–protein interaction (DPI) network of gambir’s bioactive compounds targeted to keloid-related target proteins is shown in Figure 2B. The pink dots represent the keloid protein targets, the orange dots represent gambir bioactive compounds, and the connecting lines represent the compound–target interactions. Gambiriin A1, Procyanidin B1, Procyanidin B3, Gambiriin A2, Epicatechin gallate, Gambiriin A3, Gambiriin C, Isogambirine, Procyanidin B2, and Procyanidin B4 are the top 10 bioactive compounds of gambir with high degree values resulting from the DPI network analysis (Appendix A).

### 2.4. Gene Ontology (GO) and Pathway Enrichment Analysis

GO enrichment analysis of the 11 target proteins in the intersection area was performed using the DAVID database. The result discovered 111 biological processes (BPs), 12 cellular components (CCs), and 17 molecular functions (MFs). Table 3 shows the sorted BPs, CCs, and MFs by degree of significance, *p* < 0.05. The BP of the gambir compound–keloid–target is significantly enriched in the regulation of cell proliferation, transcription, protein phosphorylation, cell migration, apoptotic process, signal transduction, peptidyl-serine phosphorylation, and protein kinase B signaling. On the other hand, the MF is mainly enriched in identical protein and enzyme binding, growth factor and cytokine activity, protein (serine/threonine/tyrosine) kinase binding and activity, and protein homodimerization activity. The CC is primarily enriched in cytoplasm, nucleus, extracellular space, extracellular matrix, extracellular region, platelet alpha granule lumen, cell surface, collagen trimer, and secretory granule.

The pathway analysis obtained 211 hits from REACTOME and 146 hits from KEGG, respectively. The first five pathways were screened and sorted according to the adjusted *p* < 0.05, which most likely shared comparable signaling pathways. The top five signaling pathways discovered from REACTOME are signaling by interleukin (IL; IL-4 and IL-13), cytokine signaling in the immune system, transcriptional regulation of transcription factors or growth factors. Meanwhile, the AGE_RAGE signaling pathway in diabetic complication, pathway in cancer, HIF-1 signaling pathway, proteoglycan in cancer, FOXO signaling pathway are the top signaling pathways discovered from KEGG. The pathway prediction of the target proteins using KEGG is illustrated in Figure 3. This analysis showed results corresponding to the GO biological process and molecular function analyses. This indicates that the bioactive compounds of *Uncaria gambir* might treat keloid by acting on these pathways.

### 2.5. Molecular-Docking Analysis

Molecular docking was used to validate the interaction between the gambir compounds and targets. The top 10 gambir bioactive compounds with high degree values were docked with TGFβ1, AKT1, and MMP1. Some compounds had lower binding energy compared with TAC as the positive control, as shown in Table 4 and visualized in Figure 4. The lower the binding energy, the more stable the binding between the ligand and the protein [25]. It was assumed that the compounds may affect the keloid pathomechanism through these target proteins. Gambiriin C and TGFβ1, Isogambirine and AKT1, and Procyanidin B1 and MMP1 are the compound–protein complexes with the highest binding energy.

## 3. Discussion

Keloid is defined as a skin fibrosis-associated disorder that is characterized by excessive growth of fibroblasts, leading to aberrant accumulation of extracellular matrix (ECM) [2]. Herbal-based therapy is expected as an alternative treatment with low toxicity and side effects in treating keloid. *Uncaria gambir* (Hunter) Roxb. is an original herbal medicine from Indonesia that has potential as an antifibrosis agent. This study first identified the bioactive compounds of gambir related to keloid target proteins through an epistatic interaction and molecular simulation approach. Network pharmacology is one of the branches of epistatic study that is considered the next paradigm in discovering the pharmacology of certain herbal compounds and their relationship to target genes of the disease. The relationship between herbs, disease, and molecular targets is discovered using a network basis that represents the action of the compound in the disease pathomechanism. Therefore, this method is considered the ideal approach for studying an herbal-based medicine that has multiple compounds and multiple targets [26,27].

This study found that there are 10 bioactive compounds of gambir with high degree values (cut-off degree value ≥ 5). Molecular-docking validation was conducted on these bioactive compounds and target proteins (TGFβ1, AKT1, and MMP1). Gambiriin C and TGFβ1, Isogambirine and AKT1, and Procyanidin B1 and MMP1 were identified as the compound–protein complexes with the highest binding energy (Table 4 and Figure 4). Therefore, Gambiriin C, Isogambirine, and Procyanidin B1 were proposed to be the most promising gambir compounds to treat keloid disease. All these compounds are classified as flavonoids, particularly chalcane-flavan dimers [28]. Flavonoids have a number of medicinal benefits, including antioxidant and anti-inflammatory properties [29]. In addition, these bioactive compounds have approved drug-likeness, pharmacokinetics, and toxicity properties as a drug candidate. Gambir’s bioactive compounds show adverse properties to ideal categories, particularly molecular weight. In term of the pharmacokinetics properties, most of the compounds are highly absorbed in GI tracts and inhibit CYP3A4 enzyme. Most of the compounds are categorized in class 4 to 6 predicted toxicity classes, meaning that most of the compounds have middle to low toxic properties.

The keloid-related proteins were represented in the PPI network produced from the datamining of two databases. DisGeNet and GeneCards are sophisticated search engines that integrate and standardize data about disease-associated genes from multiple sources, including the scientific literature [30,31]. The other network was constructed by merging the previous PPI network and gambir–target network, resulting in the DPI network (Figure 2B). The networks were then analyzed to identify the most impactful proteins in the network based on three important values. The first value is the degree of a node; this referred to the number of edges between nodes in the network. The greater the degree, the more important the nodes in the network. The second is the BC; this represents the signal probability passed through a node. The higher the BC value, the more important the node. The third is the CC value, which reflects the network tightness. The tighter the network, the higher the efficiency [8]. Based on the analysis, 11 targets were identified as the main targets of gambir bioactive compounds related to keloid disease: EP300, AKT1, STAT3, TGFB1, VEGFA, EGFR, CCND1, IGF1R, TIMP1, MMP1, and PDGFA. Thus, it is speculated that the bioactive compounds of gambir may have pharmacological activity in keloid through these targets.

GO enrichment analysis of the 11 potential protein targets was performed to clarify the role of the gambir-targeted proteins in relation to the gene function. The biological process showed that regulation of cell proliferation, transcription, protein phosphorylation, cell migration, apoptotic process, signal transduction, peptidyl-serine phosphorylation, and protein kinase B signaling are the main aspects whereby gambir may play a crucial role in dealing with keloids. In line with this, the molecular function was mainly enriched in identical protein and enzyme binding, growth factor and cytokine activity, protein (serine/threonine/tyrosine) kinase activity, and protein homodimerization activity. Interestingly, the enrichment result shared the same main characteristic of keloid disease. The main characteristic of keloid is over-synthesis and accumulation of extracellular matrix (ECM) due to effusive fibroblast proliferation and differentiation. The fibroblast is the key player in tissue fibrosis, including keloid. The activation of keloid fibroblasts is stimulated by the paracrine effect of cytokines and growth factor in the keloid’s surrounding micro-environment [2,4,32]. Dysregulation of cytokines and growth factor at any stage of the wound-healing process may lead to prolonged inflammation as a favorable condition for keloid formation [33].

In parallel with the cellular component enrichment, the protein targets were primarily enriched in cytoplasm, nucleus, extracellular space, extracellular matrix, extracellular region, platelet alpha granule lumen, cell surface, collagen trimer, and secretory granule. The highly proliferated fibroblast and its activation into a myofibroblast leads to excessive ECM synthesis and deposition, particularly collagen [34]. Keloid scars have a 20-fold increase in collagen production compared with normal skin [3]. Overproduction of ECM is closely related to prolonged inflammation in keloid pathogenesis. Keloid formation is driven by multiple cytokines, growth factors, and transcription factors [2,35]. The study by Nangole et al. [36] revealed that the high expression of proinflammatory cytokines in keloid impacts the sensitivity and increases the production of collagen on keloid fibroblasts. This is in accordance with the result of the pathway enrichment analysis from REACTOME that showed signaling by cytokine, transcription factor, and growth factor were predicted as the main pathways of gambir on keloid. Some of the cytokines and growth factors counted as main players in tissue fibrosis are interleukin-4 (IL-4), IL-13, PDGF, and TGF-β1 [37].

The closest pathway prediction was generated from KEGG, which showed TGFB1 as one of the frontline proteins in the pathway (Figure 3). TGF-β is the key activator of fibroblasts in response to fibrosis. TGF-β could be released from epithelial or endothelial cells, resident fibroblasts, platelets, and macrophages [38]. TGF-β initiates a downstream signaling cascade through activation of the Smad2/3 protein and then forms a complex protein with Smad4. This complex protein migrates to the nucleus as the regulator of ECM organization, fibroblast proliferation, and differentiation [38,39]. From the same group of growth factors, PDGF regulates fibroblast activity through the phosphatidylinositol3-kinase–AKT (PI3K/Akt) pathway [40]. The activation of the catalytic subunit of PI3K undergoes enzymatic function to catalyze phosphorylation of phosphatidylinositol 4,5-bisphosphate to phosphatidylinositol 3,4,5-trisphosphate. These triphosphate molecules activate the Akt/PKB protein to induce cell proliferation and inhibit apoptosis [41]. Most of growth factor signaling pathways are categorized as protein kinase receptors with homodimerization activity, including PDGF, EGF, VEGF, and TGF-β receptor [42]. This is in accordance with the result of the molecular function enrichment analysis.

The TGF-β pathway is associated with epithelial mesenchymal transition (EMT) activation in the keloid pathomechanism. This is a process whereby epithelial cells lose adhesion and acquire expression of a mesenchymal component and manifest the increase of cell migration. EMT plays an important role in fibroblasts differentiation into myofibroblasts. There are four common sources of myofibroblasts: fibrocytes, resident fibroblasts, endothelial cells, and epithelial cells [37]. EMT may be induced by epidermal growth factor (EGF) and EGFR via activation of Smad2/3 [43]. Another study proved that the EGF/EFGR signaling pathway promotes EMT mediated by STAT3 [44,45]. Research by Lee et al. [46] on STAT3 found signaling was the most significantly enriched Gene Ontology term in keloid fibroblast. Along with the activation of EMT, fibroblast differentiation and activation impact the imbalance of ECM remodeling.

Abnormal ECM reconstruction during wound healing contributes to keloid formation. Unlike normal scars, a decreased level of MMP1 and an increased level of TIMP1 are found in keloids that are related to tissue fibrosis [47]. MMP1 is one of the type of collagenase that catalyzes the cleave of types I and III collagens. Collagen I and III are highly expressed in keloid tissue [48,49]. Meanwhile, TIMP1 is the endogenous inhibitor of MMP1. The imbalance between the expression of MMP and TIMP leads to excessive collagen I and III synthesis and deposition in fibrotic tissue like keloid [47]. The expression of MMP and TIMP is regulated by TGF-β. The inhibition of TGF-β and the subsequent downstream signaling contribute to a more balanced MMP1/TIMP1 ratio and may be a promising therapeutic approach for keloid treatments [50].

This recent study revealed the multicomponent, multitarget, and multipathway mechanism of *Uncaria gambir* bioactive compounds against keloid through network interaction and molecular-docking simulation. EP300, AKT1, STAT3, TGFB1, VEGFA, EGFR, CCND1, IGFR, TIMP, MMP, and PDGFA are defined as the core proteins targeted by gambir’s bioactive compounds through profibrotic cytokines and growth factor signaling pathways. On the other hand, other signaling pathways should not be ignored. Therefore, the other pathway enrichment analysis result is worthy of further study. In addition, further experimental research is required to analyze and validate the mechanism of gambir in keloid identified in this study.

## 4. Materials and Methods

### 4.1. Mining Databases of Gambir Bioactive Components and Target Protein

The bioactive compounds of *Uncaria gambir* were obtained from a literature study. The structure and canonical simplified molecular-input line-entry (SMILE) were accessed from PubChem (https://pubchem.ncbi.nlm.nih.gov) between 4–8 January 2023. The bioactive compounds that were predicted to be targeted to keloid-related target proteins were listed for further analysis. The data were then submitted to SwissADME (http://www.swissadme.ch/) on 14 July 2023 to predict the absorption, distribution, metabolism, and excretion (ADME) parameters, pharmacokinetics properties, and drug-likeness nature. The Mw, HBA and HBD and Log P were utilized as parameters to consider the drug-likeness assessment. The Log P value was presented as the consensus the partition coefficient between n-octanol and water (LogP_o/w_), which was generated from SwissADME [51]. The prediction of the toxicity profile of the compounds was analyzed using a toxicity database, Protox II (https://tox-new.charite.de/protox_II/) (accessed on 16 July 2023). Furthermore, SWISS target prediction (http://www.swisstargetprediction.ch/), searching for target proteins corresponding to each molecule with only human (*Homo sapiens*) genes, was set as data retrieval. The list of proteins was aligned using UniProt (http://www.uniprot.org/uploadlists/) to eliminate double-listed proteins (accessed on 20 June 2023).

The protein targets related to keloid were identified in the Gene Cards (https://www.genecards.org/) and DisGeNet (https://www.disgenet.org/) databases using “Keloid” as the keyword in the search engine column (accessed on 13 February 2023). The identified proteins were filtered to only *Homo sapiens* genes. We utilized a Venn script (https://bioinfogp.cnb.csic.es/tools/venny/) to identify keloid-related protein targets at the intersection of the databases [52]. This tool was also used for the gambir bioactive compound targets, keloid-related targets, and protein targets related to keloid with high degree values intersection lists.

### 4.2. Construction of Protein–Protein Interaction (PPI) and Drug–Protein Interaction (DPI) Network

The PPI and DPI networks were constructed, visualized, and analyzed using Cytoscape 3.9.1 software on 15 February 2023. The PPI network was constructed from the keloid-related protein list. The proteins were uploaded to the String database (https://string-db.org/) on 20 February 2023, with 0.98 as the high confidence score for protein interaction and *Homo sapiens* set as the only organism. STRING suggests a threshold of 0.15 to be considered as low confidence, 0.40 as medium confidence, and 0.90 as high confidence. The higher confidence score of the interaction reflects higher supporting evidence and reduced false positives. On the other hand, the higher confidence may reduce the sensitivity and potentially exclude valid interactions inadvertently [23,53].

Afterwards, the network was analyzed to discover the top rank of significance proteins based on the degree, betweenness centrality (BC), and closeness centrality (CC) scores. The network was then visualized using GeneMANIA (http://genemania.org). This website allows users to find genes likely to share functions based on the interaction [54]. The second network is drug–protein interaction (DPI). The network was created to visualize and analyze the interactions of the bioactive compounds of gambir with keloid-related target proteins. Furthermore, the networks were merged to determine a new network of compound–targets of gambir targeted to keloid disease. The screening of the key bioactive compounds was also subjected to this step. The bioactive compounds with high degree values were considered the key important compounds [55].

### 4.3. Gene Annotation (GO) and Pathway Enrichment Analyses

The core protein targets from the previous step were uploaded to perform enrichment analysis for Gene Ontology (GO), biological process (BP), cellular component (CC), and molecular function (MF), as well as the pathway. All of these were conducted to describe the gene set mechanism of action through the Database for Annotation, Visualization, and Integrated Discovery (DAVID, https://david.ncifcrf.gov/), REACTOME (https://reactome.org/), and Kyoto Encyclopedia of Genes and Genomes (KEGG) (https://www.genome.jp/kegg/). The list of genes was based on the adjusted *p* value of less than 0.05 as the cut-off value that indicates statistical significance [56,57]. All of these analysis were accessed between 20 June 2023 and 7 July 2023.

### 4.4. Molecular Docking

The interactions between the gambir bioactive compounds and target proteins were validated using molecular-docking analysis. The top 10 gambir bioactive compounds with HDVs were searched for and downloaded in SDF format for the 2D structure in PubChem (https://pubchem.ncbi.nlm.nih.gov/) on 25 July 2023. The 2D structures were then prepared and saved as pdb format using MarvinSketch 23.11 software. TGFB1, AKT1, and MMP1 were selected as the target proteins, considering that these proteins were in the up, middle, and downstream of the KEGG pathway prediction. Triamcinolone acetonide (TAC) was used as a positive control since this chemical has been widely used for keloid treatment due to its anti-inflammatory properties [58]. The pdb formats of the 3D structure were downloaded from the RCSB PDB (https://www.rcsb.org/) database on 28 July 2023. Moreover, 3KFD, 4GV1, and 1HFC were the PDB ID for TGFβ1, AKT1, and MMP1, respectively. All the protein and ligand structures were saved as pdbqt formats. Molecular docking between the gambir bioactive compounds and target proteins was conducted using AutoDockTools 1.5.6 software on 9 August 2023. The ligands were set as a rigid structure and positioned at the central of cavity, which was docked by the native ligand corresponding to each protein model. Afterwards, the grid box was adjusted for the x-, y-, and z-axis for each protein models and the grid box with the lowest binding energy was selected for further analysis. The Lamarckian Genetic Algorithm was employed for running the molecular-docking simulation. The absolute value of the docking score was identified as the binding affinity and stability between the compound and target. Discovery Studio 4.5. was used to visualize the result of the molecular-docking analyses on 15 August 2023 [59].

## 5. Conclusions

In conclusion, this study first predicted the interactions of the bioactive compounds of gambir with keloid target proteins through an epistatic interaction and molecular simulation approach. EP300, AKT1, STAT3, TGFB1, VEGFA, EGFR, CCND1, IGFR, TIMP, MMP, and PDGFA were defined as the core proteins targeted by gambir’s bioactive compounds with high degree values. Gambiriin C and TGFβ1, Isogambirine and AKT1, and Procyanidin B1 and MMP1 were identified as the compound–protein complexes with the highest binding energy. GO enrichment and pathway analysis found that gambir’s bioactive compounds may act on keloid-related target proteins to regulate cell proliferation, migration, transcription, and signal transduction activity via cytokine, transcription, and growth factor signaling pathways. This is proposed to be correlated to the dysregulation of cytokines and growth factors in prolonged inflammation as favorable conditions for keloid formation. This study provides a reference for potential targets and pathways to explain the molecular mechanism of gambir’s bioactive compounds against keloid disease.

## Figures and Tables

**Figure 1 pharmaceuticals-17-00662-f001:**
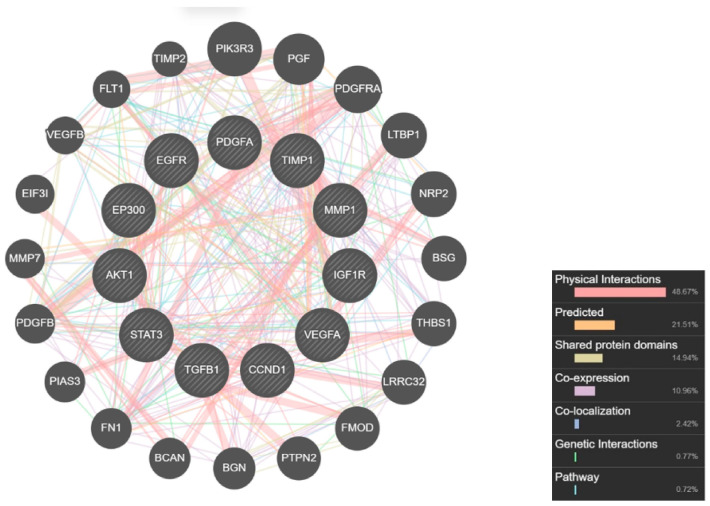
PPI network for the candidate targets. Black node indicates target protein and the correlation is represented as connecting colors.

**Figure 2 pharmaceuticals-17-00662-f002:**
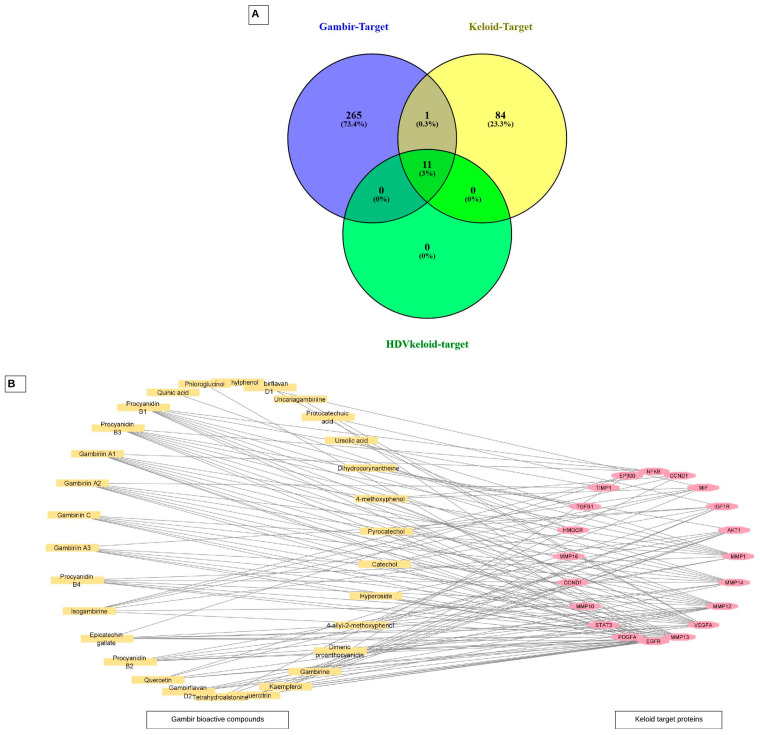
(**A**) Intersection Venn diagram between the predicted target proteins of gambir bioactive compounds, keloid-related target proteins, and high degree of connection values of keloid-related target proteins. (**B**) Compound–target network of gambir bioactive compounds targeted to keloid disease.

**Figure 3 pharmaceuticals-17-00662-f003:**
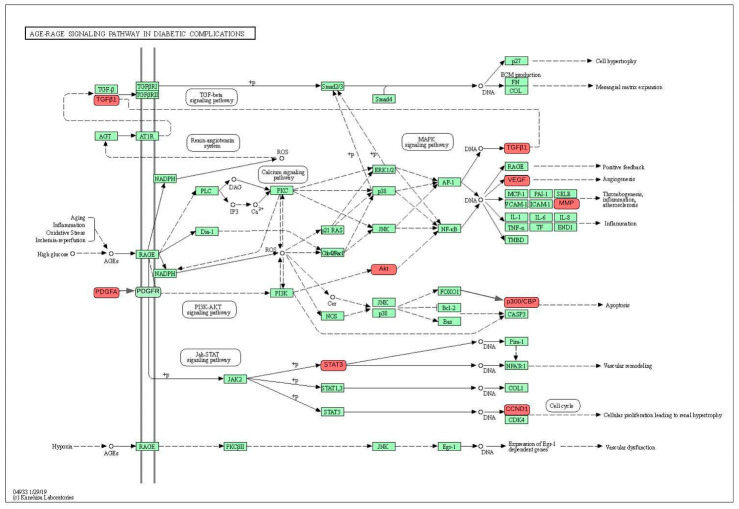
Pathway prediction of the target proteins using KEGG. The red nodes are potential target proteins of *Uncaria gambir*, while the green nodes are relevant targets in the pathway.

**Figure 4 pharmaceuticals-17-00662-f004:**
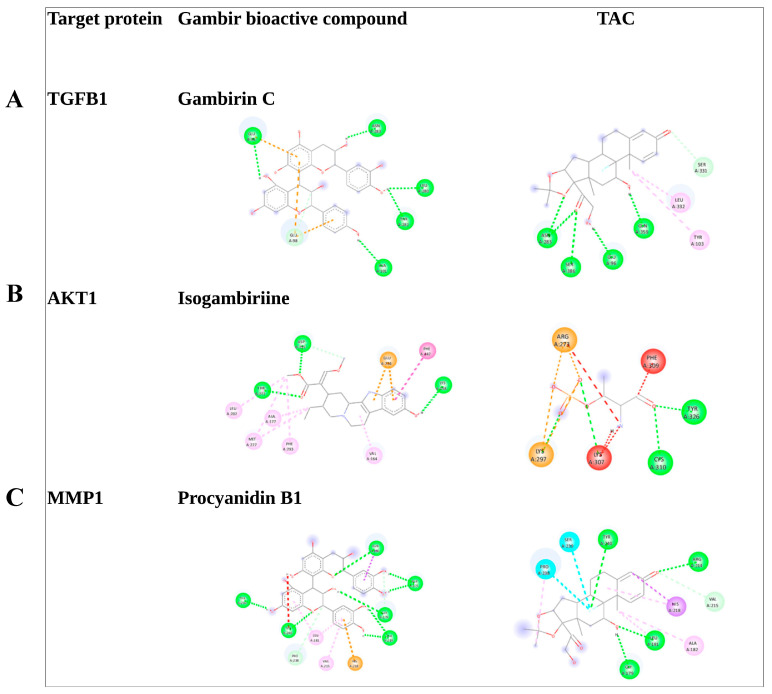
Three best molecular docking results: TGFB1 and Gambiriin C (**A**), AKT1 and Isogambirine (**B**), and MMP1 and ProcyanidinB1 (**C**).

**Table 1 pharmaceuticals-17-00662-t001:** The drug-likeness nature, pharmacokinetics properties, and toxicity of *Uncaria gambir* bioactive compounds.

No	Molecule	Drug-Likeness				Pharmacokinetics		Predicted Toxicity Class
		Mw	Log P	HBA	HBD	GI Abs	CYP Inhibitor	
**1**	Catechol	110.11	0.94	2	2	High	No	3
**2**	Gambiriin C	562.52	1.84	11	9	Low	CYP3A4	5
**3**	Gambiriin A1	580.54	1.69	12	11	Low	CYP3A4	5
**4**	Gambiriin A3	580.54	1.45	12	11	Low	CYP3A4	5
**5**	Procyanidin B3	578.52	1.53	12	10	Low	CYP3A4	5
**6**	Procyanidin B1	578.52	1.53	12	10	Low	CYP3A4	5
**7**	Gambiriin A2	580.54	1.69	12	11	Low	CYP3A4	5
**8**	Procyanidin B2	578.52	1.53	12	10	Low	CYP3A4	5
**9**	Procyanidin B4	578.52	1.53	12	10	Low	CYP3A4	5
**10**	Ursolic acid	456.70	5.88	3	2	Low	No	4
**11**	Quercetin	302.24	1.23	7	5	High	CYP2D6CYP3A4CYP3A2	3
**12**	Quinic acid	192.17	1.75	6	5	Low	No	6
**13**	Kaempferol	286.24	1.58	6	4	High	CYP2D6CYP3A4CYP3A2	5
**14**	Protocatechuic acid	154.12	0.65	4	3	High	CYP3A4	4
**15**	Epicatechin gallate	442.37	1.25	10	7	Low	No	4
**16**	Dimeric proanthocyanidin	576.50	1.61	12	9	Low	CYP3A4	5
**17**	Gambirflavan D1	562.52	2.29	11	9	Low	CYP2C9 CYP3A4	5
**18**	Gambirflavan D2	562.52	2.29	11	9	Low	CYP2C9CYP3A4	5
**19**	Hyperoside	464.38	−0.25	12	8	Low	No	5
**20**	Isoquercitrin	464.38	−0.25	12	8	Low	No	5
**21**	Pyrocatechol	110.11	0.97	2	2	High	CYP3A4	3
**22**	Phloroglucinol	126.11	0.45	3	3	High	CYP3A4	3
**23**	4-methoxyphenol	124.14	1.41	2	1	High	No	4
**24**	4-allyl-2-methoxyphenol	164.20	2.25	2	1	High	CYP1A2	4
**25**	Gambirine	384.47	2.85	5	2	High	CYP2D6	3
**26**	Isogambirine	384.50	2.82	5	2	High	CYP2D6	3
**27**	Tetrahydroalstonine	352.43	2.67	4	1	High	CYP2D6	3
**28**	Dihydrocorynantheine	368.47	3.22	4	1	High	CYP2D6 CYP3A4	3
**29**	Uncariagambiriine	620.65	3.61	9	6	Low	CYP2C9	4

**Table 2 pharmaceuticals-17-00662-t002:** Top 11 keloid-related core protein targets in the PPI network.

Gene	Gene Full Name	BC	CC	Degree
*EP300*	E1A-binding protein p300	0.19	0.43	12
*AKT1*	AKT serine/threonine kinase 1	0.12	0.44	10
*STAT3*	Signal transducer and activator of transcription 3	0.26	0.39	10
*TGFB1*	Transforming growth factor beta 1	0.11	0.44	8
*VEGFA*	Vascular endothelial growth factor A	0.17	0.43	7
*EGFR*	Epidermal growth factor receptor	0.18	0.44	6
*CCND1*	Cyclin D1	0.05	0.41	4
*IGF1R*	Insulin-like growth factor 1 receptor	0.02	0.39	3
*TIMP1*	TIMP metallopeptidase inhibitor 1	0.11	0.39	2
*MMP1*	Matrix metallopeptidase 1	0.14	0.33	1
*PDGFA*	Platelet-derived growth factor alpha	0.07	0.31	1

**Table 3 pharmaceuticals-17-00662-t003:** GO analysis of core target proteins.

Category	Gene Function	Count	*p*-Value
BP	Positive regulation of cell proliferation	6	1.9 × 10^−6^
BP	Positive regulation of transcription from RNA polymerase II promoter	6	1.0 × 10^−4^
BP	Positive regulation of protein phosphorylation	5	1.7 × 10^−6^
BP	Positive regulation of cell migration	5	3.9 × 10^−6^
BP	Negative regulation of apoptotic process	5	5.8 × 10^−5^
BP	Positive regulation of transcription, DNA-templated	5	2.1 × 10^−4^
BP	Negative regulation of transcription from RNA polymerase II promoter	5	6.8 × 10^−4^
BP	Signal transduction	5	1.8 × 10^−3^
BP	Positive regulation of peptidyl-serine phosphorylation	4	8.5 × 10^−6^
BP	Positive regulation of protein kinase B signaling	4	2.3 × 10^−5^
MF	Identical protein binding	6	5.4 × 10^−4^
MF	Enzyme binding	4	6.1 × 10^−4^
MF	Growth factor activity	3	2.7 × 10^−3^
MF	Cytokine activity	3	3.5 × 10^−3^
MF	Protein kinase activity	3	1.3 × 10^−2^
MF	Protein serine/threonine/tyrosine kinase activity	3	1.7 × 10^−2^
MF	Protein kinase binding	3	2.4 × 10^−2^
MF	Protein homodimerization activity	3	4.5 × 10^−2^
CC	Cytoplasm	8	2.5 × 10^−3^
CC	Nucleus	7	2.2 × 10^−2^
CC	Extracellular space	5	7.1 × 10^−3^
CC	Extracellular matrix	4	1.5 × 10^−4^
CC	Extracellular region	4	6.1 × 10^−2^
CC	Platelet alpha granule lumen	3	3.7 × 10^−4^
CC	Cell surface	3	3.1 × 10^−2^
CC	Collagen trimer	2	4.0 × 10^−2^
CC	Secretory granule	2	4.2 × 10^−2^

**Table 4 pharmaceuticals-17-00662-t004:** Docking scores of *Uncaria gambir* bioactive compounds with target proteins.

Compound/Affinity (kcal/mol)	TGFβ1	AKT1	MMP1
Gambiriin A1	−4.47	−7.43	−5.32
Procyanidin B1	−6.76	−3.66	−9.54
Procyanidin B3	−6.56	−6.52	−7.49
Procyanidin B2	−6.03	−5.98	−8.38
Gambiriin A2	−5.53	−5.27	−5.99
Epicatechin gallate	−5.18	−7.79	−9.48
Gambiriin A3	−5.45	−7.68	−6.24
Gambiriin C	−6.90	−7.99	−8.46
Isogambirine	−6.02	−8.68	−8.44
Procyanidin B4	−6.62	−8.47	−8.84
TAC	−6.19	−8.88	−6.69

## Data Availability

All data are available in this manuscript.

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
