# Peer review of "Profibrotic Inflammatory Cytokines and Growth Factors Are Predicted as the Key Targets of Uncaria gambir (Hunter) Roxb. in Keloids: An Epistatic and Molecular Simulation Approach"

_pharmaceuticals, 2024, doi:10.3390/ph17060662_

Round 1

Reviewer 1 Report

Comments and Suggestions for Authors

The work presented here comprises of computational and predictive report on gambir active compounds as possible drug candidates against keloids. I have a few observations and comments to make. 

What did the authors mean by the term high degree in l.197 sentence, read as "with high degree value" ? 

What is the reference for the work mentioned in l.289-291? How that report compare with the findings of the authors? What are commonality and disagreements between these two works?

How did the 3D-structures of the ligands prepared? What are the parameters used in docking protocol? was the docking experiment a bind docking or site directed docking?

The empirical nature of druglikeness needs to be understood by the authors. If simply go by Lipinski's rule, most of the molecules have far too many H-bond donor acceptor to be considered as drug target. How did the authors generate druglikeness? What are the origin of logP values? How toxicity is predicted? These information should be presented in the material and method section. 

Comments on the Quality of English Language

l. 18: Please edit. It is difficult to understand in the present form.

l. 52-54: Please edit to.... Traditional herbal medicine possesses the characteristics of being multicomponent, multifunction, 53

and multitarget; is of low toxicity and side effects, in general.

l.65 replace "proved" with "have shown"

l.108 Full form of BC and CC scores in the first appearance of these abbreviations

l. 311 What is the significance of the bold italicized sentence?

l. 358 Please remove the superscript 24

l. 358 Please replace "format" with "formats"

Author Response

Response to Reviewer 1 Comments

1. Summary

Thank you very much for your feedback on this manuscript. We appreciate the time you have dedicated to reviewing our work. Please find the detailed responses below and the corresponding revisions in track changes in the re-submitted files.

2. Questions for General Evaluation

Reviewer’s Evaluation

Response and Revisions

Does the introduction provide sufficient background and include all relevant references?

Yes

Are all the cited references relevant to the research?

Can be improved

Is the research design appropriate?

Can be improved

Are the methods adequately described?

Must be improved

Are the results clearly presented?

Can be improved

Are the conclusions supported by the results?

Can be improved

3. Point-by-point response to Comments and Suggestions for Authors

Comments 1: What did the authors mean by the term high degree in l.197 sentence, read as "with high degree value" ?

Response 1: The “degree” (l.202) corresponds to the degree value resulted from the analysis of interaction between Gambir bioactive compounds and keloid-related target proteins by using Cytoscape 3.9.1. We have added the data in Table S3 (supplementary files) to accomplish the statement.

Comments 2: What is the reference for the work mentioned in l.289-291? How that report compare with the findings of the authors? What are commonality and disagreements between these two works?

Response 2: We accordingly revised the sentence to clarify that “A recent study” (l.299) means This recent study. We are sorry for the confusion statement.

Comments 3: How did the 3D-structures of the ligands prepared? What are the parameters used in docking protocol? was the docking experiment a bind docking or site directed docking?

Response 3: Agree. We have added the ligand preparation and molecular docking protocol in l.373-381.

Comments 4: The empirical nature of druglikeness needs to be understood by the authors. If simply go by Lipinski's rule, most of the molecules have far too many H-bond donor acceptor to be considered as drug target. How did the authors generate druglikeness? What are the origin of logP values? How toxicity is predicted? These information should be presented in the material and method section.

Response 4: Agree. Some detail informations about druglikeness, Log P values, and prediction of toxicity has been supplemented in material and method section (l.317-321). In addition to Lipinski’s rule, we consider including The Mw, HBA & HBD and Log P value as the parameters to provide a more nuanced assessment of druglikeness.

4. Response to Comments on the Quality of English Language

Point 1: l. 18: Please edit. It is difficult to understand in the present form.

Response 1: Agree, we have revised the sentence (l.19) from “Keloid is characterized as the fibrotic tissue due to by high activation of fibroblast” to “Keloid is characterized as the fibrotic tissue resulting from the increase of fibroblast activity”.

Point 2: l. 52-54: Please edit to.... Traditional herbal medicine possesses the characteristics of being multicomponent, multifunction, 53 and multitarget; is of low toxicity and side effects, in general.

Response 2: Agree, we have modified the sentence as your suggestion. (l.53-56)

Point 3: l.65 replace "proved" with "have shown"

Response 3: Agree, we have changed the word as your comment. (l.67)

Point 4: l.108 Full form of BC and CC scores in the first appearance of these abbreviations

Response 4: Agree, we have revised it with full form of the abbreviation in l.111-112.

Point 5: l. 311 What is the significance of the bold italicized sentence?

Response 5: We have removed the bold italic sentence. At first, we put “Mining database of target protein related to keloid disease” as the individual subheading section. Hence, we decided to compile it with mining database of gambir bioactive compounds. (l.325-326)

Point 6: l. 358 Please remove the superscript 24

Response 6: Agree, we have removed the superscript at the l.375

Point 7: l. 358 Please replace "format" with "formats"

Response 7: Agree, we have changed the word as your comment. (l.375)

5. Additional clarifications

Dear the editor and reviewers,

Respectfully, we would like to propose the inclusion of Prof. Takashi Yashiro, MD., PhD. From Jichi Medical University (Japan) as an additional author to our manuscript. His significant contributions as collaborator of the research. We believe his expertise would enhance the equality and credibility of the work. Furthermore, we kindly request to designate Rahimi Syaidah, PhD as the corresponding author for this manuscript, in lieu of Dr. Fadilah. Thank you for your understanding.

Reviewer 2 Report

Comments and Suggestions for Authors

In this manuscript, Sri Suciati Ningsih et al. investigates the interaction of Gambir bioactive compounds with keloid target proteins using network pharmacology and a molecular simulation approach to provide a possible mechanism of action of Gambir.

The manuscript has very important qualities and is very well thought out. However, I think it has some minor and major considerations. 

Lines 88, 89, and 300 Bioactive compounds of Uncaria gambir were obtained from literature study. Need a reference

4.1. Mining database of gambir bioactive components and target protein, it is recommended to add references

Define HDV line 119

Authors are recommended to review lines 124 and 125. The pink dots represent gambir bioactive compounds, the orange dots represent protein targets, and the connecting lines represent compound-target interaction. It seems to be the other way around.

Authors are advised to read the manuscript carefully, check abbreviations, and homogenize terms.

Authors are encouraged to expand the discussion of MMP1.

Authors are encouraged to provide dates of access to databases and software.

It is recommended to put references to materials and methods 4.1.

It would be highly recommended that the data be corroborated on cell lines (RT-PCR to fibroblasts).

Author Response

Response to Reviewer 2 Comments

1. Summary

Thank you very much for taking the time to review this manuscript. Please find the detailed responses below and the corresponding revisions highlighted changes in the re-submitted files.

2. Questions for General Evaluation

Reviewer’s Evaluation

Response and Revisions

Does the introduction provide sufficient background and include all relevant references?

Yes

Are all the cited references relevant to the research?

Yes

Is the research design appropriate?

Must be improved

Are the methods adequately described?

Can be improved

Are the results clearly presented?

Can be improved

Are the conclusions supported by the results?

Yes

3. Point-by-point response to Comments and Suggestions for Authors

Comments 1: Lines 88, 89, and 300 Bioactive compounds of Uncaria gambir were obtained from literature study. Need a reference

Response 1: Agree. We have added references. (l.91)

Comments 2: 4.1. Mining database of gambir bioactive components and target protein, it is recommended to add references

Response 2: Agree. The references are the same with comments 3. (l.91)

Comments 3: Define HDV line 119

Response 3: Agree. We have defined the HDV abbreviation. (l.123)

Comments 4: Authors are recommended to review lines 124 and 125. The pink dots represent gambir bioactive compounds, the orange dots represent protein targets, and the connecting lines represent compound-target interaction. It seems to be the other way around.

Response 4: Agree. We have revised the sentences. (l.128-130)

Comments 5: Authors are advised to read the manuscript carefully, check abbreviations, and homogenize terms.

Response 5: Agree. We have modified the abbreviation terms in l.28-29, l.111-112, l.127.

Comments 6: Authors are encouraged to expand the discussion of MMP1.

Response 6: Agree. We have elaborated the discussion of MMP1 to emphasize the role of MMP1 in keloid and its relationship to TIMP1 and TGF-β in l.291-298.

Comments 7: Authors are encouraged to provide dates of access to databases and software.

Response 7: Agree. We have added the dates of access to database and software in l.312, l.315, l.325, l.328, l.363 to 364, l.370, l.376, l.379, and 387.

Comments 8: It is recommended to put references to materials and methods 4.1.

Response 8: The references are the same as comment 3 and 4. (l.90)

Comments 9: It would be highly recommended that the data be corroborated on cell lines (RT-PCR to fibroblasts).

Response 9: Agree. We will complete the analysis with qRT-PCR data on the primary culture of keloid fibroblast. Unfortunately, we wish the time expansion due to the indent time of some reagents for those experiment.

5. Additional clarifications

Dear the editor and reviewers,

Respectfully, we would like to propose the inclusion of Prof. Takashi Yashiro, MD., PhD. From Jichi Medical University (Japan) as an additional author to our manuscript. His significant contributions as collaborator of the research. We believe his expertise would enhance the equality and credibility of the work. Furthermore, we kindly request to designate Rahimi Syaidah, PhD as the corresponding author for this manuscript, in lieu of Dr. Fadilah. Thank you for your understanding.

Round 2

Reviewer 1 Report

Comments and Suggestions for Authors

Thank you for your response letter.

Reviewer 2 Report

Comments and Suggestions for Authors

The authors addressed all my suggestions